# Theoretical Study on Pressure Damage Based on Clinical Purpura during the Laser Irradiation of Port Wine Stains with Real Complex Vessels

**Hao Jia [1,2], Bin Chen [1,\*] and Dong Li [1]**

[1] State Key Laboratory of Multiphase Flow in Power Engineering, Xi'an Jiaotong University, Xi'an 710049, China; jiahao@zstu.edu.cn (H.J.); lidong@mail.xjtu.edu.cn (D.L.)

[2] State-Province Joint Engineering Lab of Fluid Transmission System Technology, Zhejiang Sci-Tech University, Hangzhou 310018, China

\* Correspondence: chenbin@mail.xjtu.edu.cn; Tel.: +86-029-8266-7326

**Abstract:** Port wine stains (PWSs) are congenital dermal vascular lesions composed of a hyperdilated vasculature. Purpura represented by local hemorrhage from water vaporization in blood during laser therapy of PWS is typically considered a clinical feedback, but with a low cure rate. In this study, light propagation and heat deposition in skin and PWSs is simulated by a tetrahedron-based Monte Carlo method fitted to curved bio-tissues. A curvature-corrected pressure damage model was established to accurately evaluate the relationship between purpura-bleeding area (rate) and laser therapy strategy for real complex vessels. Results showed that the standard deviation of Gaussian curvature of the vessel wall has negative relation with the fluence threshold of vessel rupture, but has positive relation with the effective laser fluence of vessel damage. This finding indicated the probable reason for the poor treatment of PWS, that is, considering purpura formation as a treatment end point (TEP) only leads to partial removal of vascular lesions. Instead, appropriate purpura area ratio with marked effects or rehabilitation should be adopted as TEP. The quantitative correlation between the fluence of a pulsed dye laser and the characteristics of vascular lesions can provide personalized and precise guidance for clinical treatments.

**Keywords:** port wine stains; laser therapy; curvature-corrected pressure damage model; purpura; treatment end point

## 1. Introduction

Port wine stains (PWSs) are congenital vascular birthmarks characterized by overabundant dermal capillaries or venules, and these marks occur in approximately 0.3% of newborns [1]. The effective treatment strategy of PWSs focuses on the laser irradiation of the affected portions of the skin [2,3]. In accordance with selective photothermolysis (SP) theory [4], laser irradiation can irreversibly induce thermal injury to abnormal blood vessels while minimally damaging the surrounding skin tissue.

A pulsed dye laser (PDL) with a wavelength of 585 or 595 nm is considered the golden standard for the treatment of PWS lesions [2,5]. In comparison with a low absorption efficiency and an unapparent response of the regions exposed to lasers at the near-infrared band, the strong absorption of PDL energy by hemoglobin can easily result in water vaporization in blood and local hemorrhage, which is clinically manifested as purpura [2,6,7], a real-time feedback of SP. According to more than 100 experiments, a laser-irradiated region remains perfused in the absence of purpura of the irradiated vessels. In clinical practice, lesions with purpura formation undergo discoloration 1 month after treatment, whereas lesions in a nonpurpura region remain dark. Therefore, purpura is typically considered a qualitative clinical treatment end point (TEP) and prognostic indicator for lesion removal [8].

The extent of purpura formation has not been regarded as an accurate indicator of treatment outcome. A too low or too high therapeutic dose can lead to vascular recanalization or scarring, and clinical therapy gradually becomes limited in improving the cure rate (only approximately 50%) of PWS nevus [9]. Purpura is characterized by blood spilling out of the skin and the mucosa after vascular rupture. A model involving "purpura-bleeding area (rate)–realistic vessel geometry–laser therapy strategy" should be established to discover the mechanism of vessel damage represented by purpura and improve the curative effect [10]. Nevertheless, minimal attention has been paid to this issue.

Blood coagulation and vessel rupture are regarded as thermal damages and modeled by the damage integral $\Omega$, which increases exponentially with tissue temperature [11–14]. After the temperature field is solved with the bioheat conduction equation, $\Omega$ can be obtained by calculating the integral related to temperature, and blood coagulation can be represented as high-$\Omega$ regions [15]. As a subsequent effect of blood coagulation, complete vessel constriction has also been modeled by $\Omega$, although its reasonable damage threshold should be clarified [8]. However, this single-parameter damage model considering thermal blood coagulation is inapplicable to vessel rupture, which is marked by purpura and has been improperly regarded as a type of thermal damage. In animal experiments utilizing a PDL, vessel wall rupture is accompanied with large bubble formation from the change in the water phase in blood [8,16]. Vessel rupture is caused by excessive pressure on a blood vessel wall after vaporization, not by heat from blood cells to the blood vessel wall. Pressure produced by gasification probably damages the blood vessel wall before heat transfers from blood cells to the blood vessel wall. Therefore, vessel rupture should be modeled by using a given pressure threshold and not by heat damage $\Omega$.

Zhang et al. [14] introduced a heat transfer equation with a pressure-dependent vaporization factor to predict temperature distribution, including water phase change in laser surgery in accordance with SP theory. However, the luminal pressure of the blood vessel in this model is only used to correct the temperature field and $\Omega$ rather than be regarded as an independent parameter to evaluate the vessel rupture potential, which cannot be predicted effectively in this manner. Our group [7] proposed a rupture potential index based on the wall pressure (RPIP) to estimate the vessel rupture. A double-parameter criterion comprising RPIP and temperature integral ($\Omega$) is also suggested to investigate the competition between the pressure damage (vessel rupture) and thermal damage (vasoconstriction) of the irradiated vessels. Our results show that the damage characteristics of the PDL and the infrared laser are fundamentally distinguished. In the PDL, vessels constantly reach the pressure threshold prior to the heat damage threshold, and the vessel undergoes rupture instead of complete constriction. With infrared laser, uniform energy is deposited in the vessel, thereby inducing much higher probability of constriction than hemorrhage. The superiority of PDL is revealed by verifying the rupture as a more severe damage type than the constriction because of the more drastic pressure variation at the vessel wall.

In clinical settings, the complex structure of the PWS vasculature results in an unpredictable rupture position and bleeding volume, leading to erroneous diagnosis by physicians. A numerical study on real blood vessels, such as aorta and celiac artery, has shown that the stress on the vessel wall often reaches the extremum at the local saddle point [17]. This finding indicates that the bending degree of the vessel surface significantly affects the pressure distribution of the vascular wall. Bucking and tortuosity occur frequently in the patient-specific PWS vessel structure, thereby changing the blood hydrodynamics and seriously affecting the rupture [18]. Available skin models usually adopt ideal cylinders as pathological vessels, but they do not represent real clinical lesions. Heat deposition in individual vessels by light absorption in the blood can be significantly reduced by competition from a branched or surrounding vasculature, which is also not considered by a simplified geometric model [19]. Consequently, an anatomically realistic geometric model of the vasculature should be established to accurately simulate the force on vessel walls, and the curvature of these walls should be investigated to quantitatively characterize the relationship between vascular morphology and pressure damage.

Before a theoretical study is conducted, a real vascular lesion should be geometrically reconstructed. Computer-reconstructed biopsy shows the detailed structure of vascular lesions, but it causes unnecessary injury to patients' skin [20]. Noninvasive skin imaging technologies, such as optical coherence tomography and laser speckle imaging, are reliable in distinguishing normal or diseased tissues on a large scale but have a low spatial resolution in 3D imaging of thin tissue layers, such as skin tissues [21]. As a type of in vivo imaging, a dorsal skin chamber (DSC) model [16,22–26] is effective in investigating the thermal response of blood vessels. The geometric feature of blood vessels in the DSC model is nearly parallel, thereby allowing 3D reconstruction from 2D blood vessel images to reveal the mechanism of real vascular injuries, especially vascular rupture and hemorrhage. Mu et al. [27] performed a preliminary 3D reconstruction of real blood vessels from the image processing of a DSC model through microscopic observation. However, a vessel rupture based on vessel wall curvature is not considered, leading to an unclear relationship between the bleeding area of blood vessels (purpura) and the parameters of lasers.

In the present study, a curvature-corrected pressure damage (CCPD) model was proposed to accurately predict the rupture of a real radiant vessel through 3D vascular reconstruction and establish the relationship between purpura-bleeding area (rate) and laser therapy strategy. The model was validated by comparing its results with the experimental results of blood vessels in the DSC model with different curvatures. The effects of vessel curvature, burial depth, and epidermal melanin content (i.e., human skin color) on the parameters of the 595 nm PDL treatment for PWSs were systematically studied to establish a quantitative relationship of the parameters of PDL therapy with the burial depth and curvature of vascular lesions and the melanin content of the epidermis. This study provided a guide on the clinical treatment of PWS.

## 2. Materials and Methods

### 2.1. Problem Description and Basic Assumptions

Human skin consists of two main layers, i.e., the superficial epidermal layer and the deep dermal layer. PWSs result from the malformation of a significant amount of blood vessels buried within the dermis. In the treatment of vascular lesions, laser energy is mainly absorbed by hemoglobin in blood. However, epidermal melanin is another chromophore for light absorption, and its content influences the risk of epidermal injury. According to SP theory [4], laser irradiation should maximally induce an irreversible injury to abnormal blood vessels while minimizing the risk of injury in the surrounding dermis and epidermis. The major assumptions in this work are as follows:

(1) The epidermis of a rat was removed in the DSC model for microscopic observation [12,13]. For convenience, the epidermal layer was excluded from the simulation to validate the theoretical model by the in vivo results from the DSC model. Afterward, the whole skin model comprising the epidermis and the dermis was adopted to study the effects of epidermal melanin content and vessel depth on laser fluence. The surfaces of the epidermis and the dermis were considered planar.

(2) The tetrahedron-based Monte Carlo (TMC) method was employed to simulate light propagation and energy deposition because a tetrahedron mesh is body fitted for a complex vessel shape [28]. The optical properties of different tissues are assumed to be constant and uniform within a given component. The following parameters were disregarded during the short laser pulse duration: changes in the density, heat capacity, and thermal conductivity of skin tissues; metabolic heat generation; and biochemical reactions.

(3) The epidermis and the dermis were treated as solid phases because of the low water content [14], and only the energy equation was involved. Thermal damage represented by temperature was used to characterize the safety of the epidermis and the dermis.

(4) The purpura area was assumed to be equal to the bleeding area in the DSC model and directly proportional to the volume fraction of blood vessels in the lesion under the same treatment parameters in accordance with the approximate relationship between the clinical treatment and

the animal experiment. The purpura image in clinical settings was analyzed to provide the appropriate area ratio of purpura for setting the laser parameters in simulation.

## 2.2. Governing Equations

The complete two-layer skin model contained the epidermis, the dermis, and the blood vessels (Figure 1). A 1.5 mm × 1.5 mm × 1.0 mm computational domain of the skin with a reconstructed blood vessel buried at a depth of 0.5 mm was used. The thicknesses of the epidermis and the dermis were set respectively as $H_e = 60$ μm and $H_d = 940$ μm based on statistical average values [15]. As mentioned in Section 2.1, in validating the damage model, the reconstructed blood vessels in accordance with the DSC model were embedded in the skin without the epidermis.

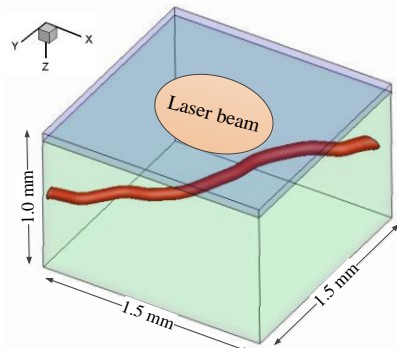

**Figure 1.** Computational domain of the two-layer skin model with a reconstructed blood vessel.

In the present work, light propagation and energy deposition in the skin tissue were simulated by using our in-house TMC code [28] through which the body-fitted material grid of the tetrahedra was generated by assigning a value representing a tissue type. No interface tetrahedral cells existed, thereby avoiding the error of photon reflection/refraction in the voxel-based Monte Carlo method [29]. Photon propagations in TMC were expressed as probability distributions that describe the step size of photon movement between sites of photon–tissue interaction. Once a photon has reached an interaction site, a fraction of the photon weight was absorbed in the local tetrahedron. The "random walk" of photon could be simulated repeatedly until a sufficient quantity of photons was absorbed in tetrahedral cells to establish reliable trends of energy deposition. In this work, the laser beam was a flat top profile with a diameter of 1.0 mm. $10^8$ photons were launched at $z = 0$ μm with their $x$ and $y$ determined by the random number restricted in the range of the beam. The directional cosines ($\mu_x$, $\mu_y$, $\mu_z$) were initialized to (0, 0, 1) and the photon weight is initialized to 1.

Energy deposition in different cells determines the heat source and thus strongly influences mass and heat flow in tissues. The following numerical formulation was implemented in the finite-volume-based CFD package FLUENT 17.0 (Ansys Inc., Canonsburg, U.S.A.) to simulate blood flow and heat transfer in vessels. The tetrahedral grid used in CFD computation was the same as that in TMC, and the energy deposition computed by TMC was treated as heat source to the governing equations. Considering that blood is an incompressible Newtonian fluid, we can write a single set of governing equations, including continuity, momentum, and energy equations in the portion of the vessel, as follows:

$$\frac{\partial \rho}{\partial t} + \nabla \cdot (\rho \mathbf{u}) = 0, \tag{1}$$

$$\frac{\partial(\rho \mathbf{u})}{\partial t} + \nabla \cdot (\rho \mathbf{u}\mathbf{u}) = -\nabla p + \nabla \cdot \left[\mu\left(\nabla \mathbf{u} + \nabla \mathbf{u}^T\right)\right] + \rho \mathbf{g} + \mathbf{F}, \tag{2}$$

$$\frac{\partial(\rho c_p T)}{\partial t} + \nabla \cdot (\rho c_p \mathbf{u}\, T) = \nabla \cdot (k\, \nabla T) + S_L + S_E, \tag{3}$$

where **F** is the surface tension, $S_L$ is the energy source computed by TMC through laser pulse, and $S_E$ is the energy source term related to evaporation or condensation.

By contrast, the epidermis and the dermis were treated as a solid phase, and only heat transfer caused by laser heating was involved:

$$\frac{\partial(\rho c_p T)}{\partial t} + \nabla \cdot (\rho c_p \mathbf{u}\, T) = \nabla \cdot (k\, \nabla T) + S_L. \tag{4}$$

The liquid–vapor mass transfer caused by laser heating (evaporation and condensation) is governed by the vapor transport equation [7], and the inbuilt VOF module in FLUENT 17.0 was adopted to capture the interface when bubbles formed in the blood.

### 2.3. Boundary Conditions

In laser dermatologic surgery, cryogen spray cooling (CSC) is applied to avoid unwanted damage from epidermal burning due to the melanin absorption of light energy. In this study, the refrigerant R134a was sprayed to the epidermis before laser pulses were administered to protect the epidermis. The spray duration was $t_c = 100$ ms, and the spray distance was 30 mm [30]. After CSC was conducted ($t > t_c$), the natural convection of air was considered the boundary condition. The standard convection condition was used to describe the cooling effect of CSC on the skin surface ($z = 0$):

$$k_e \left.\frac{\partial T}{\partial z}\right|_{z=0} = \begin{cases} h_c \cdot (T - T_c), & t \le t_c \\ h^*(r^*, \tau) \cdot (T - T_a), & t > t_c \end{cases}, \tag{5}$$

where $k_e$ is the thermal conductivity of the epidermis, $T_a = 25\ ^\circ$C is the ambient temperature, and $h_a = 10$ W·m$^{-2}$·K$^{-1}$ is the convective heat transfer coefficient of air. $T_c$ of $-48.5\ ^\circ$C [30] is the temperature of the refrigerant liquid film. The dimensionless surface heat transfer coefficient $h^*(r^*, \tau)$ based on a large number of experimental data was expressed as follows [30]:

$$\begin{aligned} h^*(r^*, \tau) &= \begin{cases} h_o^*(\tau) & 0 \le r^* \le 0.4 \\ \frac{5(1-r^*)}{3h_o^*(\tau)} & 0.4 < r^* \le 1.0 \end{cases} & \tau \le 1.0 \\ h^*(r^*, \tau) &= \begin{cases} h_o^*(\tau) & 0 \le r^* \le 0.2(\tau+1) \\ h_o^*(\tau) + \frac{(5r^* - \tau - 1)[0.09(\tau-1) - h_o^*(\tau)]}{(4-\tau)} & 0.2(\tau+1) < r^* \le 1.0 \end{cases} & 1.0 < \tau < 4.0 \\ h^*(r^*, \tau) &= h_o^*(\tau) \qquad 0 \le r^* \le 1.0 \quad \tau \ge 4.0 \end{aligned}, \tag{6}$$

where nondimensional time $\tau = t/t_{max}$, and nondimensional space $r^* = r/r_{spray}$; the expression of the dimensionless convective heat transfer coefficient at the center of refrigerant $h_0^*(\tau)$ is presented as follows [30]:

$$h_o^*(\tau) = \begin{cases} \tau & \tau \le 1.0 \\ 1.0 - 0.35(\tau - 1) & 1.0 < \tau \le 3.0 \\ 0.3 - 0.02(\tau - 3) & 3.0 < \tau \le 8.0 \\ 0.2 - 0.0125(\tau - 8) & \tau > 8.0 \end{cases}. \tag{7}$$

Adiabatic conditions were adopted for the side and bottom boundaries. The initial skin temperature was assumed to be 35 $^\circ$C, and the inlet velocity of the blood was set as $1 \times 10^{-3}$ m/s [31].

### 2.4. CCPD Model

The morphological characteristics of a real curved blood vessel can be represented by Gaussian curvature. The two eigenvalues $k_1$ and $k_2$ of Weingarten transformation at any point on a surface are called the principal curvatures of surface $S$ at that point [32]. Gaussian curvature $K_G$ is defined as the product of the principle curvatures [32] as follows:

$$K_G = k_1 k_2. \tag{8}$$

In this work, the vessel wall was discretized by a triangular mesh. Meyer et al. [32] obtained the discrete operator of Gaussian curvature by using the geometric parameters of the one-ring neighborhood mesh:

$$K_G(\mathbf{x}_i) = (2\pi - \sum_{j=1}^{n} \theta_j)/A_{vor},$$ (9)

where $j$ is the interior angle of the $j$th triangle at the vertex $x_i$, $n$ is the number of triangles around the vertex, and $A_{vor}$ is the area of Voronoi region [33,34] in the one-ring neighborhood of the vertex (Figure 2):

$$A_{vor} = \frac{1}{8} \sum_{j \in N_1(i)} (\cot \alpha_j + \cot \beta_j)\|\mathbf{x_i} - \mathbf{x_j}\|,$$ (10)

where $N_1(i)$ is the point set of all vertices (except $i$ itself) in the one-ring neighborhood of vertex $i$, and $\alpha_j$ and $\beta_j$ are the two angles opposite to the edge in the two triangles sharing the edge ($\mathbf{x_i}$, $\mathbf{x_j}$) as depicted in Figure 2.

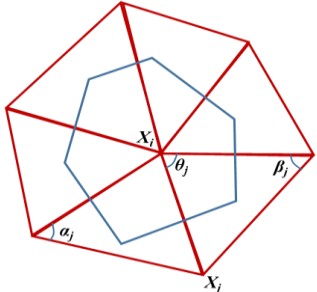

**Figure 2.** Finite-volume region on a one-ring triangulated surface of $\mathbf{x_i}$ by using Voronoi cells. For the point inside each adjacent triangle, the surface area with the circumcenters is recognized as the local Voronoi cell (region surrounded by the blue lines).

The computation error of Gaussian curvature could be controlled to less than 1% by using this algorithm [32].

After the Gaussian curvature was precisely calculated, virtual pressure $P_{Ga}$ caused by Gaussian curvature was introduced to consider the effect of the anatomical geometry of blood vessels on rupture, and this parameter was added to the wall pressure $P$:

$$P' = P + P_{Ga}$$ (11)

where $P'$ is the $K_G$-corrected pressure on the vessel wall. Statistical analysis [35] showed that $K_G$ on the vessel wall follows a negative linear relationship with the additional wall stress caused by curvature (Pearson's r = −0.437). The pressure on the vessel wall at the moment of laser irradiation was considered proportional to the stress according to our previous work [7]. Therefore, $P_{Ga}$ under laser irradiation can be assumed to be linear with the Gaussian curvature on the vessel wall:

$$P_{Ga} = A \cdot K_G + B$$ (12)

where $A$ and $B$ are constants.

In this study, an ideal vessel had a cylindrical surface with a Gaussian curvature of zero. Thus, the intercept $B$ should be zero when the cylindrical surface was considered a standard without additional pressure. The Gaussian curvatures of convex and concave surfaces were positive and negative, respectively, and the pressure capacity of the concave surface was less than that of the convex surface. Therefore, the expression of the virtual pressure caused by the curvature could be written as:

$$P_{Ga} = -P/P_{w,max} \cdot K_G \tag{13}$$

where $P_{w,max}$ is the maximum wall pressure of blood vessels. The model based on $P'$ is named as the CCPD model, which differs from the pressure damage (PD) model based on $P$ [7]. The pressure threshold to vessel damage was chosen as 2500 Pa through analogy analysis on pressure thresholds of rupture on different vessels in our previous work [7]. The rupture area of the blood vessel ($S_{br}$) could be determined by the area with $P'$ higher than 2500 Pa. The validation of the CCPD model through rupture area under a given threshold of rupture pressure is discussed in the next section.

The region of purpura or bleeding had a recognizable contour that could be captured by Image Pro Plus. Koster et al. [36] and Tanghetti et al. [37] conducted an immediate histopathological analysis of patients with PWS after laser treatment and found that $S_{br}$ follows a linear relationship with the area of purpura ($S_{pu}$). Therefore, the functional relationship between $S_{br}$ and $S_{pu}$ could be obtained by linearly fitting between the computation and the animal experiment. The linear fit according to our damage model is also discussed in the next section.

## 3. Validation of CCPD Model

### 3.1. 3D Reconstruction of Real Blood Vessels

Considering the approximate parallelism of the blood vessel with the observation plane in the DSC model, the blood vessel axis was regarded on a horizontal plane. The dorsal blood vessels of male Sprague–Dawley rats weighing 100–120 g were used as a target to be irradiated. To make the observation chamber, the dorsal skin was lifted and fixed between two symmetrical titanium frames containing a central annular window (Ø = 15 mm). The entire epidermal, dermal, and subcutaneous connective tissues of one side of the skin in the central annular window were carefully removed to expose the vascular structure (Figure 3). The fascia on the exposed site was carefully removed to improve the imaging. The rats were placed on a specially engineered microscope translator stage containing three pins that interlock with holes in the implanted optical chamber so as to minimize motional disturbances by the breathing of the anesthetized rats during imaging. Proper targeted blood vessel was oriented by mounting the xyz-translator stage. Then, the skin images were captured with a CCD camera (DP71, Olympus Corp., Tokyo, Japan) connected to the microscope (10 × objective, NA = 0.30, BX51, Olympus Corp.) at a resolution of 1360 × 1024. The images were simply processed and displayed by cellSens standard software. All of the procedures involving the animal experiments were approved by the Institutional Animal Care and Use Committees of Xi'an Jiaotong University.

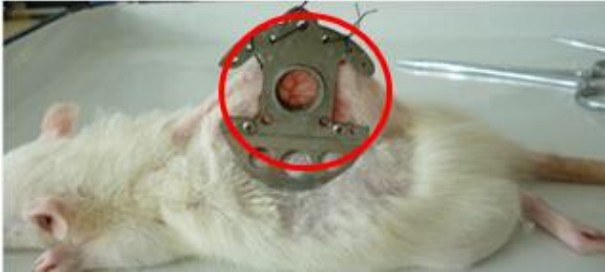

**Figure 3.** Dorsal skin chamber model.

The procedure of 3D reconstruction of real blood vessels is shown in Figure 4. After a snapshot of the DSC model was taken (Figure 4a), the binary image of the main blood vessels was transformed (Figure 4b), and the skeleton and contour lines of the blood vessels were extracted from the single 2D image (Figure 4c). After a smooth and loft operation (Figure 4d), the 3D blood vessel model was generated on the basis of the skeleton line and the contour [27] (Figure 4e). Only one image instead of multislices was used for reconstruction; thus, complex operations, such as identifying and assembling

blood vessel regions between slices, were avoided, and the reconstruction steps were greatly simplified. The generated 3D geometric model was imported into GAMBIT and discretized by tetrahedral meshes.

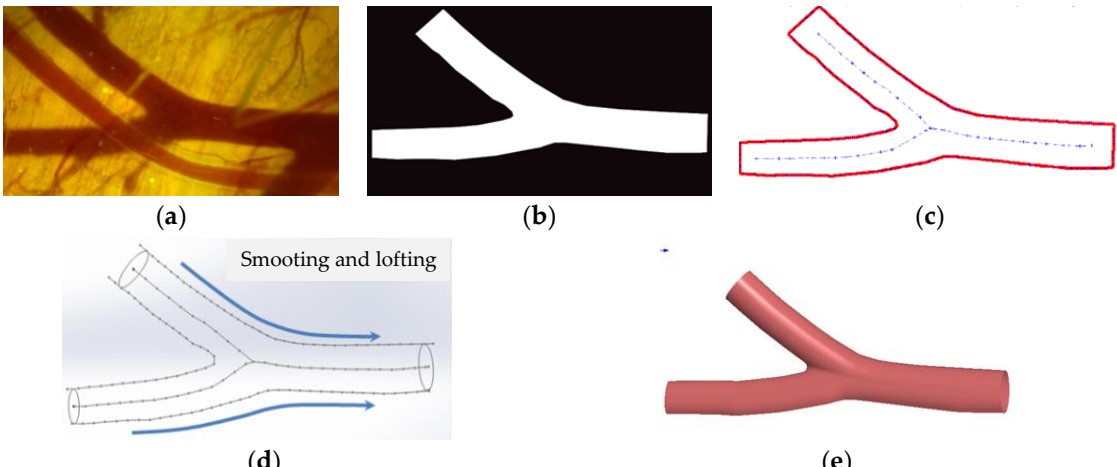

(**a**)  (**b**)  (**c**)

(**d**)  (**e**)

**Figure 4.** Procedure of the 3D reconstruction of real blood vessels. (**a**) Taking a photo of the DSC model; (**b**) extracting the main blood vessels and transforming them to a binary image; (**c**) extracting the contour and skeleton lines; (**d**) smoothing the line and outputting it in a vector form; (**e**) conducting 3D angiogenesis.

*3.2. Grid Independence Test*

According to the characteristic of blood vessel geometry in the DSC model[5], the reconstructed blood vessels buried at a depth of 0.5 mm with an average diameter of 100 ± 10 μm were embedded in the skin model to verify the rupture damage mode and validate the CCPD model (Figure 5). As mentioned in Section 2.2, only the dermal layer and the blood vessel in it were considered and compared with those in the in vivo experiment. The skin depth was 1 mm, and the cross section of the skin was 1.5 mm × 1.5 mm–2 mm × 2 mm, depending on the size of each blood vessel. A 595 nm PDL with a pulse width of 6 ms and a spot diameter of 1 mm was used in the in vivo experiment and the numerical simulation. The optical and thermal properties of dermis and blood corresponding to the wavelength of 595 nm are presented in Table 1.

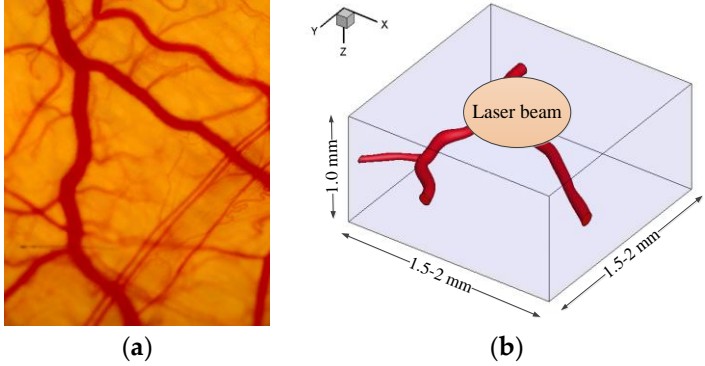

(**a**)  (**b**)

**Figure 5.** Geometric skin model with a 3D reconstructed blood vessel. (**a**) Vessel image of the dorsal skin chamber model (the main vessel of reconstruction is indicated by the yellow arrow). (**b**) Skin model with a 3D reconstructed vessel.

**Table 1.** Optical and hydrodynamic properties of dermis and blood [7].

| | Property | Dermis | Blood |
|---|---|---|---|
| Optical properties | Absorption coefficient, $\mu_a$/cm$^{-1}$ | 20 | 49.3 |
| | Scattering coefficient, $\mu_s$/cm$^{-1}$ | 460 | 466 |
| | Anisotropy index, $g$ | 0.8 | 0.995 |
| | Refractive index, $n$ | 1.37 | 1.33 |
| Thermal properties | Density, $\rho$/kg·m$^{-3}$ | 1090 | 1060 |
| | Thermal conductivity, $k$/kW·m$^{-1}$·K$^{-1}$ | 0.41 | 0.55 |
| | Specific heat, $c$/J·kg$^{-1}$·K$^{-1}$ | 3500 | 3600 |

Six different tetrahedron grids with average interval sizes of 20, 15, 12.5, 10, 7.5, and 5 μm in the vessel region were tested for the reconstructed blood vessel (Figure 5) to check the grid independence of the present algorithm. Figure 6 shows the local Gaussian curvature at the vessel bifurcation. Figure 7 depicts the Gaussian curvature variation with different grid intervals of points A and B on the vessel wall (Figure 6). The average deviation in GC between the results with cellular sizes of 7.5 and 5 μm at the two points was less than 1.1%. The grid independence test showed that 7.5 μm was the appropriate size for temperature and pressure computation. Therefore, the grid with a cell size of 7.5 μm was adopted in the rest of our simulations.

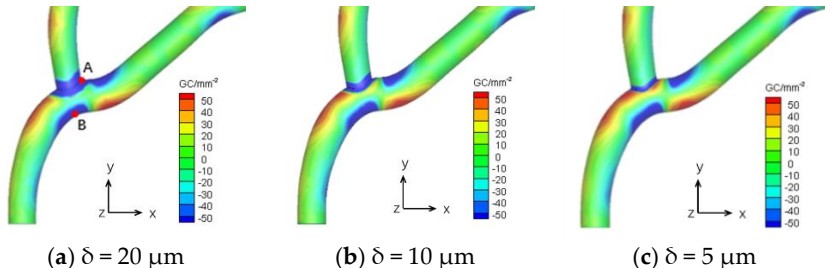

(**a**) δ = 20 μm　　　　(**b**) δ = 10 μm　　　　(**c**) δ = 5 μm

**Figure 6.** Local Gaussian curvature obtained from the mesh with different intervals.

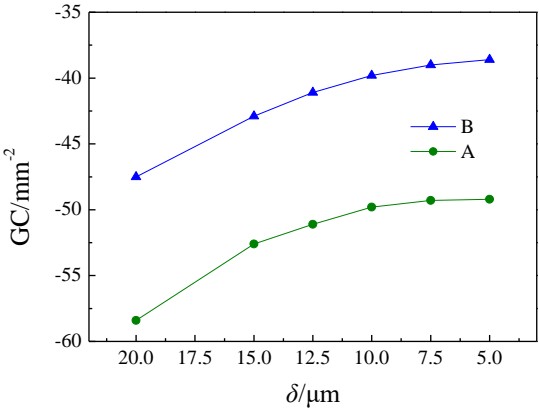

**Figure 7.** Influence of the mesh on the calculation of the Gaussian curvature on the bifurcated vessels.

*3.3. Validation of the CCPD Model*

Both the skin model and the damage model are considered as important factors in the computation. In this section, three computational models are made by combination from either of the two blood vessel geometries and the two damage models. For simplicity, the three models were referred to as models A, B, and C, which are a PD model with an ideal vessel (a straight cylinder), a PD model with a real vessel, and a CCPD model with a real vessel. With $P$ = 2500 Pa as the pressure damage threshold [7], the location and area of the vessel rupture in the three models were computed and compared with those of the experiments.

Based on our computation, the standard deviation of the Gaussian curvature (StdGC) of a single vessel ranged from 10 to 80. In this study, the blood vessels with a diameter of 100 ± 10 μm were divided into three groups in terms of their StdGC: small (0 ≤ StdGC < 25), medium (25 ≤ StdGC < 50), and large (StdGC ≥ 50). For each group, one typical vessel was chosen to analyze pressure distribution. Figure 8 shows the experimental snapshots and the reconstructed blood vessels. For the convenience of model comparison, the position of the ideal cylindrical vessel was adjusted to coincide with the reconstructed vessel (Figure 9).

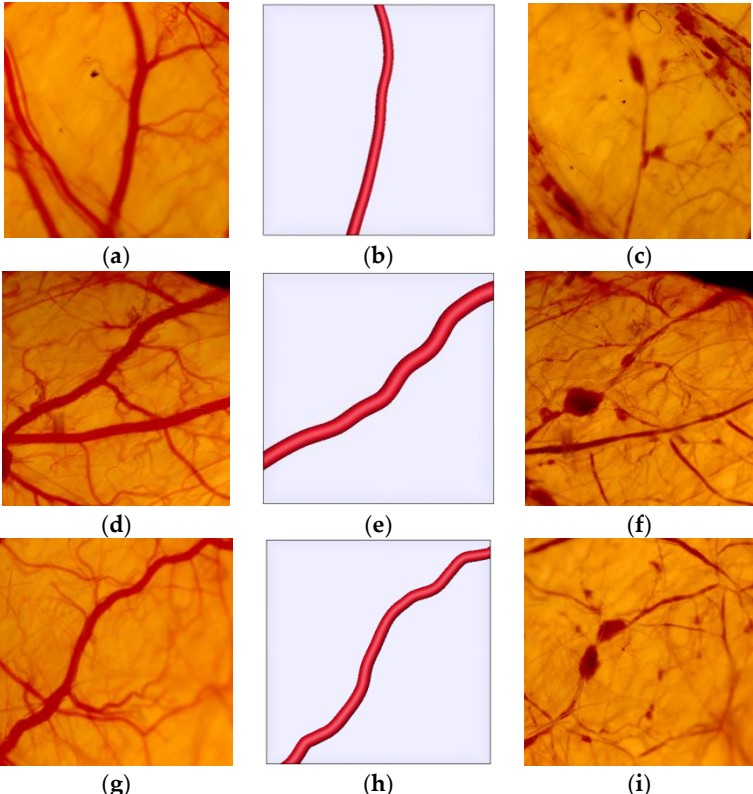

**Figure 8.** Responses of blood vessels irradiated with a 595 nm laser with a pulse width of 6 ms (yellow arrow refers to the irradiated reconstructed vessel) (**a**) Before irradiation (stdGC = 20.5); (**b**) reconstructed model (stdGC = 20.5); (**c**) after irradiation (stdGC = 20.5); (**d**) before irradiation (stdGC = 41.3); (**e**) reconstructed model (stdGC = 41.3); (**f**) after irradiation (stdGC = 41.3); (**g**) before irradiation (stdGC = 64.5); (**h**) reconstructed model (stdGC = 64.5); and (**i**) after irradiation (stdGC = 64.5).

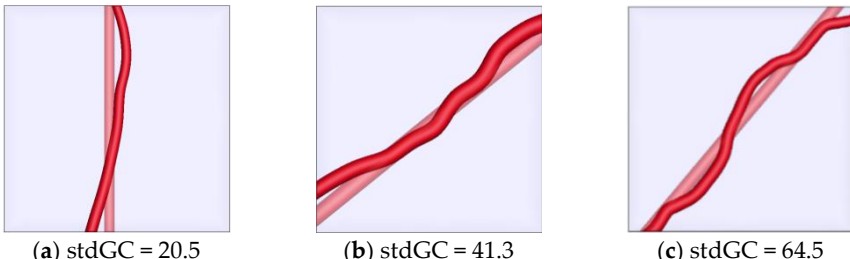

(**a**) stdGC = 20.5   (**b**) stdGC = 41.3   (**c**) stdGC = 64.5

**Figure 9.** Reconstructed vessel models with different curvatures and corresponding cylindrical vessel models (the dark red vessel is the 3D reconstructed vessel, and the light red vessel is the corresponding cylindrical vessel). The position of the ideal vessel was adjusted to coincide with the reconstructed vessel.

Figures 10–12 show the wall pressure distribution of the three models. Model A has the largest rupture area, whereas model B has the smallest rupture area because the tortuosity of the real vessel disturbs the blood flow, which restricts the growth of large bubbles or breaks them into small bubbles.

Therefore, the contact area between the region of high pressure (gas phase) and the blood vessel wall reduces, leading to a small rupture area. The rupture area of the real vessel in model C is between those of models A and B, and the rupture site agrees with the large Gaussian curvature spot on the wall surface, such as the saddle point. Figure 8c,f,i illustrate that the high-pressure position and area in model C (the present CCPD model) are more consistent with those of rupture and bleeding in the animal experiments than those in the two other models. This finding confirms the validity and feasibility of the new model in predicting vessel hemorrhage.

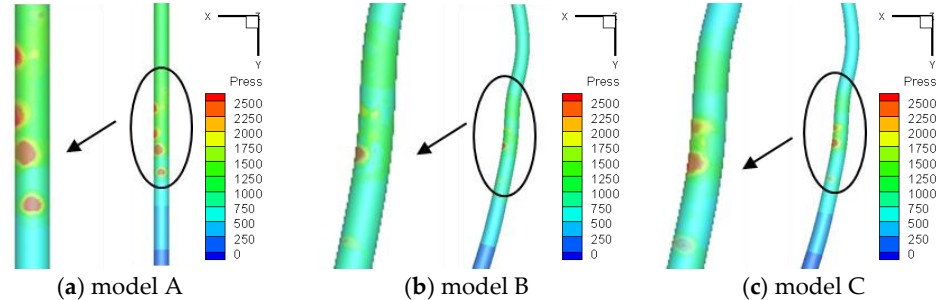

(**a**) model A  (**b**) model B  (**c**) model C

**Figure 10.** Pressure distribution on the wall of small curvature (stdGC = 20.5 mm$^{-2}$) vessels after laser irradiation ($E$ = 11 J/cm$^2$, $t_p$ = 6 ms).

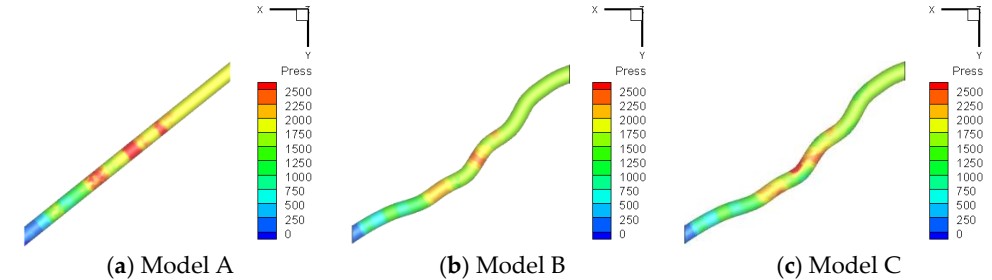

(**a**) Model A  (**b**) Model B  (**c**) Model C

**Figure 11.** Pressure distribution on the wall of the median curvature (stdGC = 41.3 mm$^{-2}$) vessels after laser irradiation ($E$ = 12 J/cm$^2$, $t_p$ = 6 ms).

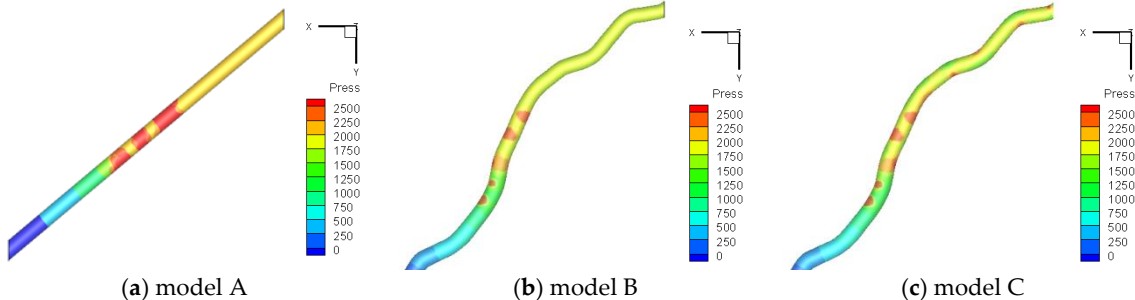

(**a**) model A  (**b**) model B  (**c**) model C

**Figure 12.** Pressure distribution on the wall of the median curvature (stdGC = 64.5 mm$^{-2}$) vessels after laser irradiation ($E$ = 13 J/cm$^2$, $t_p$ = 6 ms).

Figure 13 shows the relationship between $S_{br}$ obtained by the three models and $S_{pu}$ measured by the experiment. For a certain laser fluence, models A and B respectively show the largest and smallest rupture areas, whereas model C has moderate-sized rupture area.

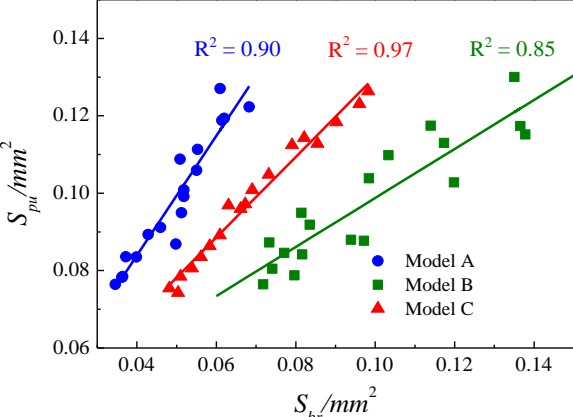

**Figure 13.** Relationship between $S_{br}$ predicted by three damage models and $S_{pu}$ measured by the experiments under different incident energy densities.

As shown in Figure 13, the CCPD model (model C) predicts a more significant linear relationship between $S_{br}$ and $S_{pu}$, which is in accordance with the histopathological analysis of patients with PWS [36,37]. After comprehensive consideration with our analysis in Figures 11–13, the CCPD model is the best predictive model. The linear relationship between $S_{br}$ and $S_{pu}$ by model C fitted in Figure 13 is expressed as:

$$S_{pu} = 1.01 S_{br} + 0.0279. \tag{14}$$

## 4. Results and Discussion

In this section, the effects of different vessel curvatures, burial depths, and epidermal melanin contents (i.e., human skin color) on the parameters of the PDL treatment of PWSs were systematically studied to determine the reasonable area ratio of the vascular purpura obtained from clinical observations. The depths of blood vessels in our simulation were set as 0.25, 0.375, and 0.5 mm corresponding to the superficially, mid-, and deeply buried blood vessels, respectively. Vessel models were selected as those with small, medium, and large StdGC, as shown in Figure 8. Laser pulse duration and laser spot diameter were fixed at $t_p = 6$ ms and $D = 1$ mm, respectively.

The melanin volume fractions ($f_m$) in the epidermis were set as 1%, 3%, 5%, 7%, and 9% corresponding to people with white, light yellow, dark yellow, brown, and black skin, respectively [38]. Assuming that melanin is evenly distributed in the epidermis, we present the optical and thermal parameters of the epidermis containing melanin in Table 2. The effect of melanin content on the absorption coefficient of the epidermis was introduced in the form of volume fraction. The same optical and thermal properties of the vessel and the dermis in Table 1 were adopted.

**Table 2.** Optical and thermal properties of the epidermis irradiated by a 595 nm pulsed dye laser [39].

| Property | Epidermis (No Melanin) | Melanin | Epidermis with $f_m$ Melanin |
|---|---|---|---|
| Absorption coefficient, $\mu_a$/cm$^{-1}$ | 0.371 | 402.63 | $\mu_{a,m} \times f_m + \mu_{a,e} \times (1 - f_m)$ |
| Scattering coefficient, $\mu_s$/cm$^{-1}$ | 470 | —— | 470 |
| Anisotropy index, $g$ | 0.79 | —— | 0.79 |
| Refractive index, $n$ | 1.37 | —— | 1.37 |
| Density, $\rho$/kg·m$^{-3}$ | 1120 | —— | 1120 |
| Thermal conductivity, $k$/kW·m$^{-1}$·K$^{-1}$ | 0.34 | —— | 0.34 |
| Specific heat, $c$/J·kg$^{-1}$·K$^{-1}$ | 3200 | —— | 3200 |

*4.1. Energy Threshold of Vessel Hemorrhage*

The area of vessel rupture is closely related to purpura caused by vessel hemorrhage. For the convenience of statistical analysis of the degree of purpura in clinical patients during laser therapy, the area ratio of purpura (bleeding) $S_{rpu}$ is introduced as:

$$S_{rpu} = \frac{S_{pu}}{N \cdot S_{spot}},$$
(15)

where $S_{pu}$ is the total area of the purpura after laser treatment, $N$ is the total number of laser pulses, and $S_{spot}$ is the area of laser spot. For the DSC model in which the epidermis was removed, $S_{rpu}$ shows the area ratio of bleeding.

Fifty patients were treated with a 595 nm PDL in Laser Treatment Center, Department of Dermatology, Second Affiliated Hospital of Xi'an Jiaotong University to establish the relationship between $S_{rpu}$ predicted through numerical simulation and the treatment criterion based on clinical $S_{rpu}$. The study included 34 female and 16 male PWS patients, with phototypes II-IV on the Fiztpatrick scale and an average age of 28. $S_{pu}$ was measured immediately after each patient was treated (Figure 14a). $S_{spot}$ is the circular area with a spot diameter of 1 mm, and $N$ is related to the morphological characteristics of vascular lesions.

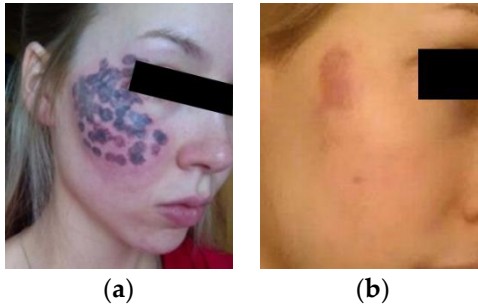

(**a**)　　　　　　　　　　　　　(**b**)

**Figure 14.** Purpura and curative effect of patients with port wine stains and treated with a 595 nm pulsed dye laser (**a**) Purpura occurs immediately after irradiation. (**b**) Return visit after 1 month of treatment (marked effect).

The reasonable range of $S_{rpu}$ after treatment could be obtained in terms of the degree of disease recovery at the return visit after 1 month of treatment (Figure 14b). If the laser fluence was too high, the bleeding area was also too large, and side effects such as erythema, scar, or pigmentation appeared after treatment. If the re-examination showed that the scar remained visible, a patient was included in the group with side effects. If no side effects were found during the return visit, patients were divided into four groups based on the clearance rate. Therefore, a total of five groups were set as follows: (a) no effect (clearance rate < 30%); (b) weak effect (30% ≤ clearance rate < 60%); (c) marked effect (60% ≤ clearance rate < 90%); (d) rehabilitation (clearance rate ≥ 90%); and (e) side effects (erythema, scar, or pigmentation).

Only a suitable laser fluence could achieve an acceptable therapeutic effect. The medical community in dermatology has reached a consensus on the effectiveness in treating PWS nevus, that is, the effective cure rate is equal to the sum of the rate of marked effect and rehabilitation. Figure 15 shows the mean and confidence interval of $S_{rpu}$ of the five groups. Mean $S_{rpu}$ increases almost uniformly as the laser fluence increases, and the curative effect changes gradually from no effect to side effects. Ideal $S_{rpu}$ should be higher than $S_{rpu}$ of marked effects (c) and lower than $S_{rpu}$ of side effects (d), that is, higher than 13.8% and lower than 25.1%, as shown in Figure 15. The volume fraction of lesions in patients with moderate PWS nevus is approximately 20%, which is twice the volume fraction of the DSC vessels; hence, the appropriate range of $S_{rpu}$ in computation based on the DSC model should be between 6.9%

and 12.6%. Laser fluence leading to 6.9% $\leq S_{rpu} \leq$ 12.6% in the DSC model experiment is defined as the effective laser fluence $E_{eff}$, i.e., the fluence threshold of appropriate vessel hemorrhage.

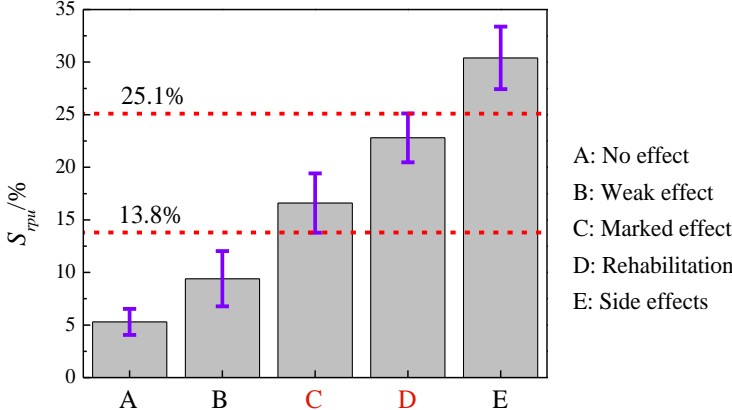

**Figure 15.** Relationship between the therapeutic effect of PWS and $S_{rpu}$ (Data source: Laser Treatment Center, Department of dermatology, Second Affiliated Hospital of Xi'an Jiaotong University).

### 4.2. Effect of Laser Transport in Skin with PWS Vessel by TMC

The effects of 595 nm PDL transport in the skin with the curved PWS vessel by TMC were studied on the two layered skin model with a domain of 1.5 mm × 1.5 mm × 1.0 mm. The thicknesses of the epidermis and the dermis were set respectively as $H_e$ = 60 μm and $H_d$ = 940 μm and a reconstructed blood vessel buried in the skin at a depth of 0.5 mm based on statistical average values [15].

Figure 16 shows laser energy deposition in skin with blood vessel of model C (shown in Section 3.3) and melanin content of 3% in epidermis ($f_m$ = 3%). Photon deposition distribution in the cross section at $y$ = 0.75 mm is shown in Figure 16a. Due to high absorbance of the blood vessel and epidermis, energy deposition concentrates in the two domains. The axis of the real blood vessel is circuitous and arbitrarily positioned, therefore its cross-section shows an ellipse not directly below the laser beam. It is apparent that the top right part of the vessel, which faces the photon initial direction, collects more energy than the bottom left part.

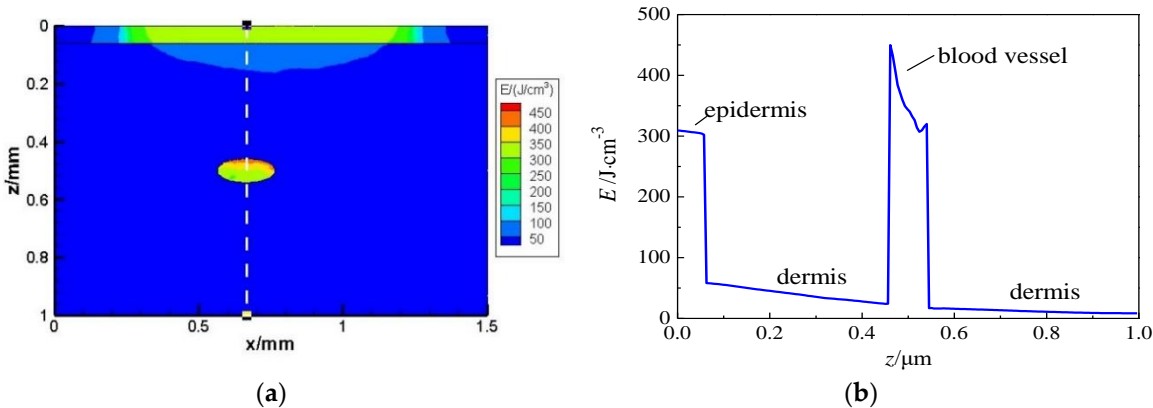

(**a**)　　　　　　　　　　　　　　　　　　　　　　　　(**b**)

**Figure 16.** Energy deposition in skin with blood vessel model C ($f_m$ = 3%, 595 nm PDL, 12 J/cm$^2$). (**a**) Cross-sectional maps of laser energy deposited at $y$ = 0.75 mm. (**b**) Line graph of energy deposition versus depth at the vessel center (shown as white dashed line in Figure 16a) for simulations of skin with single blood vessel.

Figure 16b presents the corresponding lateral energy deposition profile. The line graph includes four parts: epidermis, dermis, blood vessel and dermis (Figure 16b). Laser energy was mostly absorbed by the blood vessel, then the epidermis and least the dermis. In each of the four parts, the energy

deposition decreases with the depth except for the bottom of the blood vessel. The slight increase at the bottom of the vessel is caused by incident and deposition of the scattered light from the deep layer of the dermis. For people with the light-yellow skin ($f_m$ = 3%), the energy absorbance of the epidermis is over 300 J/cm$^2$. In this condition, whether spray cooling is necessary depends on the results of heat transfer calculation in the next Section 4.3.

### 4.3. Effect of Melanin Content in the Epidermis on Laser Fluence Selection

The effects of melanin content in the epidermis were investigated from 1% to 9% on a real vessel under 595 nm PDL irradiation. Figure 17 shows $S_{rpu}$ of a vessel buried in the median layer (0.375 mm) at $f_m$ of 1% or 5%. The laser fluence of the vessel rupture increases with the melanin content. For example, in the blood vessel of stdGC = 41.3, the bleeding threshold increases from 4 J/cm$^2$ to 6 J/cm$^2$ when $f_m$ increases from 1% to 5% because the light absorption capacity of the epidermis increases with melanin content (Table 2). Subsequently, the relevant absorption ratio of the blood vessel decreases. Therefore, a high laser fluence is required to rupture the targeted blood vessel. With an increase in laser fluence, $S_{rpu}$ of the vessel with different values of stdGC continues to increase and gradually achieves a significant effect (6.9% < $S_{rup}$ < 12.6%). However, when the laser fluence continues to increase, $S_{rup}$ rapidly exceeds the upper limit (12.6%), and the skin experiences a side effect.

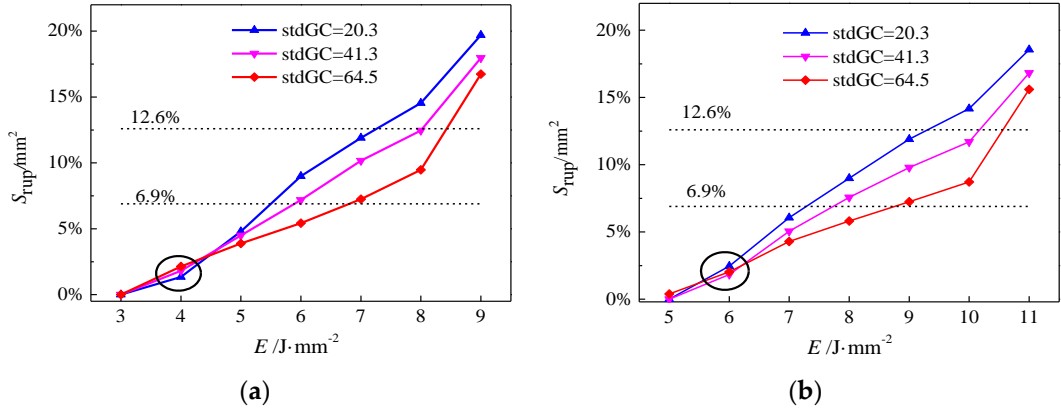

**Figure 17.** Purpura rate of blood vessels at a depth of 0.375 mm (the laser fluence of vessel rupture is marked by the black circles): (**a**) $f_m$ = 1% (**b**) $f_m$ = 5%.

For $f_m$ = 1%, three vessels with different values of stdGC simultaneously reach the rupture threshold at $E$ = 4 J/cm$^2$. Among them, the vessels with a large stdGC have a large Gaussian curvature at the saddle point of the vessel surface. As a consequence, rupture and bleeding occur easily. When the laser fluence exceeds 5 J/cm$^2$, the rupture area of the vessel with a small stdGC increases faster than that with a large stdGC because the lumen of the vessel with a small stdGC is relatively straight. As a result, the disturbance to the blood flow and the generated bubbles is weak. The bubbles grow and come in contact with the vessel wall without restriction, leading to a large high-pressure area on the vessel wall and rupture. At $f_m$ = 1%, when the laser fluence reaches 9 J/cm$^2$, the gap in the rupture area of the three vessels is sharply narrowed because high energy greatly reduces the curvature effect. Similar results are obtained at $f_m$ = 5%.

In laser dermatologic surgery, CSC with R134a is an efficient technique to avoid epidermal damage among white people because of light energy absorption by melanin. However, even with CSC protection, people with dark pigmentation may suffer from epidermal damage. Previous studies showed that thermal damage occurs when the central temperature of the epidermis $T_{ec}$ is higher than 80 °C at the end of laser irradiation [40]. Therefore, the PDL fluence threshold of epidermal injury $E_{e,R134a}$ is defined as the laser fluence leading to $T_{ec}$ = 80 °C with R134a cooling. $E_{e,R134a}$ increases with an increase in the cooling effect of cryogen. $E_{eff}$ should be higher than $E_{e,R134a}$ to ensure the safety of the epidermis in the treatment of vascular lesions.

Figure 18 shows $E_{eff}$ at different melanin contents and a vessel depth of 0.375 mm. $E_{eff}$ varies by 1–2 J/cm$^2$ with the degree of curvature regardless of $f_m$. With the present CSC technology, for white ($f_m$ = 1%) and light-yellow ($f_m$ = 3%) people, the epidermis remains undamaged when $E_{eff}$ is used. For dark-yellow people ($f_m$ = 5%), $E_{eff}$ of a large-stdGC vessel exceeds $E_{e,R134a}$. Therefore, $E_{e,R134a}$ should be improved by strengthening the cooling capacity of cryogen. For brown ($f_m$ = 7%) and black ($f_m$ = 9%) people, $E_{eff}$ is higher than $E_{e,R134a}$. The near-infrared laser instead of PDL should be used to treat darkly pigmented patients because of its weak absorption of melanin.

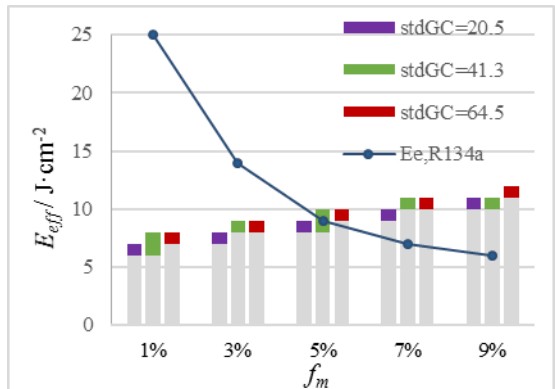

**Figure 18.** $E_{eff}$ of laser at different melanin contents with the vessel buried at a depth of 0.375 mm (marked by solid rectangles).

## 4.4. Effect of Vessel Depth on the Selection of Laser Fluence

Deep PWS vessels are usually resistant to laser surgery. In this study, the effect of vessel depth on the PDL treatment of PWS nevus was investigated. Figure 19 shows $S_{rpu}$ of a vessel buried in the superficial layer (0.25 mm) or the deep layer (0.5 mm) when $f_m$ is 3%. A shallow buried blood vessel leads to rupture and bleeding at low fluence. When $H$ decreases from 0.5 mm to 0.25 mm, the bleeding threshold decreases significantly from 8 J/cm$^2$ to 2 J/cm$^2$ because the deeper the buried vessel is, the thicker the dermis will be for light propagation to reach the vessel. In this regard, more photon energy is absorbed and scattered by the dermis, so the energy required for vessel rupture consequently increases.

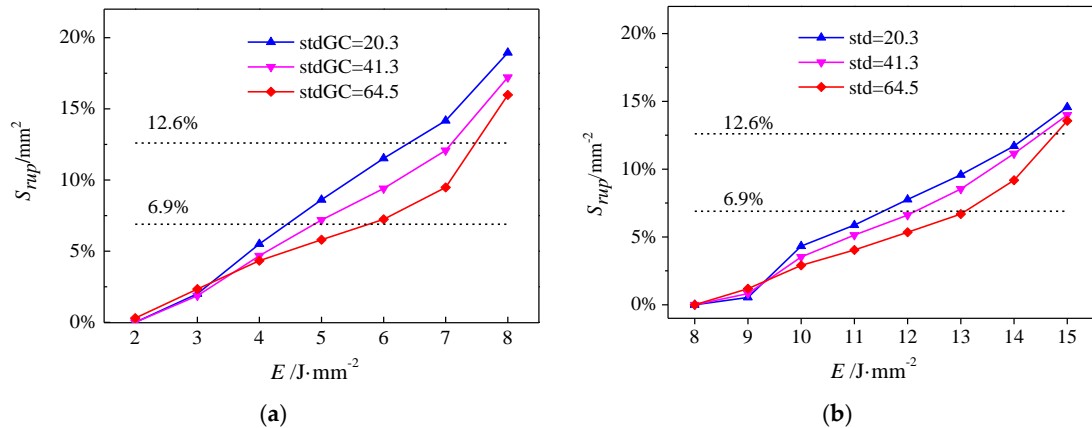

**Figure 19.** Purpura rate of blood vessels with melanin content $f_m$ = 3%. (**a**) Vessel burial depth $H$ = 0.25 mm (**b**) Vessel burial depth $H$ = 0.5 mm

At the same burial depth, the rupture thresholds and $E_{eff}$ of the vessels with different stdGC vary. For the vessel depth of $H$ = 0.25 mm and $H$ = 0.5 mm, the rupture fluence of the vessels with a large stdGC (64.5) is lower than that with a medium or small stdGC, whereas $E_{eff}$ is 1–2 J/cm$^2$ higher.

As the laser fluence increases, the vessels with a large stdGC initially rupture, subsequently experience purpura, and finally reach $E_{eff}$. The occurrence of purpura does not mean that the vessel has been properly treated. The optimal treatment parameters can only be determined by the appropriate $S_{rpu}$. Simply taking the occurrence of purpura as the TEP leads to an inadequate treatment and a high recurrence rate of vessels with a large stdGC. This observation reveals one of the reasons why clinically inexperienced physicians fail to treat many patients with PWS. Therefore, in the treatment of vessels with a large stdGC, the therapeutic effect should be determined on the basis of appropriate $S_{rpu}$ (higher than $S_{rpu}$ of marked effect and lower than $S_{rpu}$ of side effects), and the required laser fluence should be higher than that of vessels with a small stdGC.

Figures 21 and 22 depict $E_{eff}$ of laser at different melanin contents and vessel depths of 0.25 and 0.5 mm. $E_{eff}$ differs by 1–2 J/cm$^2$ with different values of stdGC. For white people ($f_m$ = 1%), superficial vessels ($H$ = 0.25 mm) can be treated reasonably without CSC, whereas deep vessels ($H$ = 0.5 mm) need CSC.

As shown in Figure 20, when the melanin content is equal to 7%, $E_{e,R134a}$ in the superficial vessel ($H$ = 0.25 mm) is lower than $E_{eff}$. Therefore, the cooling effect of the epidermis should be improved. For median ($H$ = 0.375 mm) and deep ($H$ = 0.5 mm) vessels, insufficient epidermal cooling occurs when the melanin contents are 5% and 3%, respectively. Therefore, for darkly pigmented skin, R134a cannot meet the requirement for treating median and deep blood vessels. Zhou et al. [30] found that an R134a spray can be substituted with low-boiling-point cryogen R404A (with a standard boiling point of −46.1 °C) to improve the surface cooling effect. Therefore, R404A can be used to cool the epidermis in the treatment of PWS lesions in darkly pigmented people and protect the epidermis from damage while removing vascular lesions.

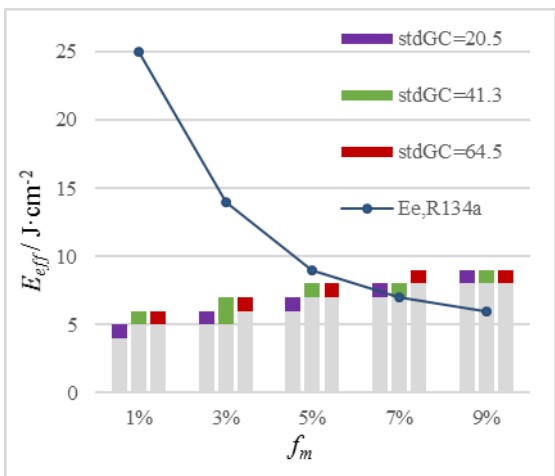

**Figure 20.** $E_{eff}$ of laser at different melanin contents in vessels buried at a depth of 0.25 mm (marked by solid rectangles).

The depth of blood vessels influences the selection range of $E_{eff}$. For a large-stdGC vessel (64.5), when the vessel depth is 0.25 or 0.375 mm, $E_{eff}$ is 2 J/cm$^2$ (shown as the height of the red rectangle in Figures 19 and 21), whereas it reduces mostly to 1 J/cm$^2$ when the vessel depth is 0.5 mm (shown as the height of the red rectangle in Figure 21). The difference in $E_{e,R134a}$ is important because its absolute value is not high, especially when vessels are buried at a depth of 0.25 mm (5–9 J/cm$^2$).

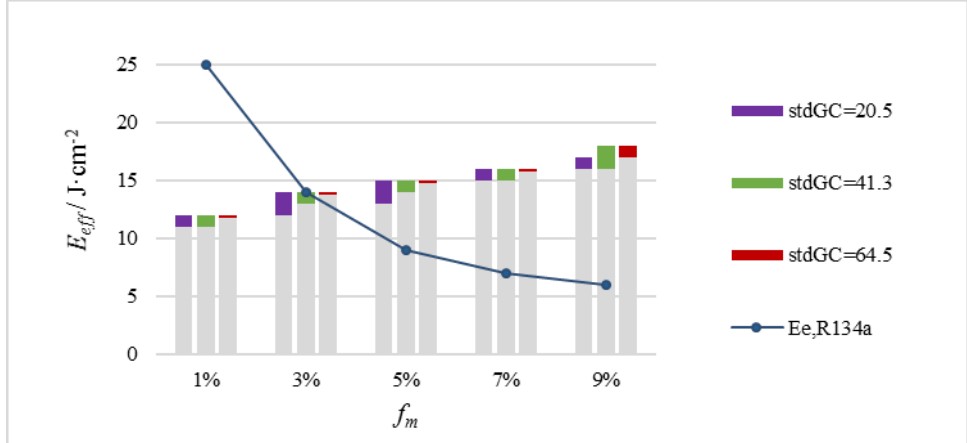

**Figure 21.** $E_{eff}$ of laser at different melanin contents in vessels buried at a depth of 0.5 mm (marked by solid rectangles).

The simulated data in Figures 18, 20 and 21 are presented in Figure 22.

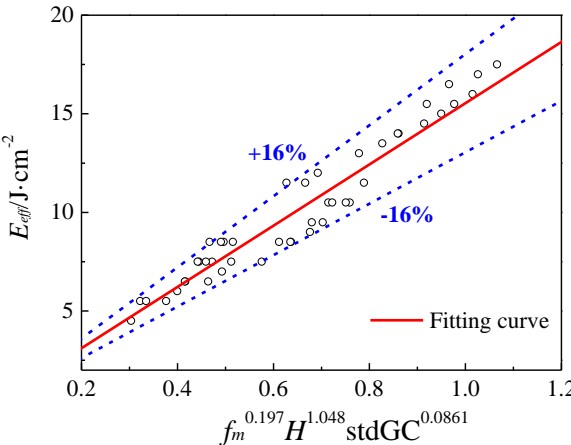

**Figure 22.** Simulation results of $E_{eff}$ with different parameters of vessel and epidermal melanin.

The relationship of effective fluence with melanin, vessel depth, and curvature can be obtained by least square fitting as follows:

$$E_{eff} = 15.536 f_m^{0.197} H^{1.048} stdGC^{0.0861}, \tag{16}$$

where vascular depth $H$ is measured in millimeters. The error between the simulation results of $E_{eff}$ and the predicted values of the fitting formula is less than ±16%, as shown in Figure 22. This correlation can predict the laser fluence threshold of vessel hemorrhage in the treatment of vascular lesions by a 595 nm PDL. The application conditions of Equation (16) are $1\% \leq f_m \leq 9\%$, $0.25$ mm $\leq H \leq 0.5$ mm, and $20.5 \leq stdGC \leq 64.5$.

In the simulation of light transport in turbid medium like skin tissues, the establishment of scattering model is important to our TMC method. For the dynamic scattering medium, special consideration should be done to build an appropriate scattering model. In the animal experiments by which our CCPD model was validated, motional disturbances by the breathing of the anesthetized rats was mostly eliminated by the fixation of translator stage and the optical chamber. But the effect of blood flow to light scattering was neglected in this work. Recently, Yang et al. [41] established an angular spectrum model to trace the field propagation during the entire optical phase conjugation process in the presence of dynamic scattering media, which can be used in the establishment of scattering model in our subsequent research. In addition, optical scattering prevents light from being focused to the target

vessel through thick dermal layer at depths greater than ~1 mm [42]. Although the vessel is buried at a depth of ~0.5 mm in our simulation, the scattering effect of the skin needs to be carefully considered. In our future work, the digital optical phase conjugation based wavefront shaping technique [42] will be introduced to improve the scattering model for simulation of light propagation in skin with deep buried blood vessel.

## 5. Conclusions

In this study, the curvature corrected pressure damage (CCPD) model was proposed to examine the relationship between bleeding area (purpura area) and laser fluence in the treatment of real PWS vessels and accurately evaluate the location and area of a vessel rupture. The main conclusions were as follows:

(1) Vascular lesions with a high stdGC are prone to hemorrhage and purpura formation but have difficulty in achieving $S_{rpu}$ to damage vascular lesions. This observation reveals one possible reason for the poor treatment of PWS in clinic, that is, purpura formation is simply regarded as the TEP, which leads to insufficient treatment and a high recurrence rate of vascular lesions with a large stdGC. An appropriate $S_{rpu}$ should be adopted as the TEP that is higher than $S_{rpu}$ of marked effects and lower than $S_{rpu}$ of side effects. With the ability to provide appropriate $S_{rpu}$, $E_{eff}$ is recommended to precisely treat vascular lesions.

(2) Our simulation results reveal the quantitative correlations of $E_{eff}$ of PDL with epidermal melanin content, vessel curvature, and depth for individualized and precise guidance on clinical treatments. To PWS patients with different melanin levels, different treatment strategies are suggested. PDL should be used carefully in treating light-yellow people ($f_m$ = 3%). Cryogen R404A with a good cooling capacity is suggested to protect the epidermis with $f_m$ = 5%. For brown ($f_m$ = 7%) and black ($f_m$ = 9%) people, the implementation of $E_{eff}$ of blood vessels in PDL severely harms the epidermis; thus, near-infrared lasers are recommended because they minimally absorb melanin.

**Author Contributions:** Conceptualization, H.J., B.C. and D.L.; Methodology, H.J.; Software, H.J.; Validation, H.J., B.C.; Formal analysis, D.L.; Investigation, H.J.; Data curation, H.J.; Writing—original draft preparation, H.J.; writing—review and editing, B.C.; supervision, B.C.; Project administration, B.C.; Funding acquisition, B.C.

**Funding:** This work was supported by the National Natural Science Foundation of China (51727811), Zhejiang Provincial Natural Science Foundation of China (LZ15E090002) and Science Foundation of Zhejiang Sci-Tech University (ZSTU) (19022101-Y, 11130231711926).

**Acknowledgments:** Thanks for the technical support of the Second Affiliated Hospital of Xi'an Jiaotong University.

**Conflicts of Interest:** The authors declare no conflict of interest.

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
