# Peer review of "Theoretical Study on Pressure Damage Based on Clinical Purpura during the Laser Irradiation of Port Wine Stains with Real Complex Vessels"

_applsci, doi:10.3390/app9245478_

Round 1

Reviewer 1 Report

In this article the curvature corrected pressure damage CCPD model was proposed to examine the relationship between bleeding area and laser fluence in the treatment of real PWS vessels

Q1: Line 17-18 Please explain the setting parameters for Monte Carlo method.

Q2: Line 162: Please explain the setting parameters for two layer skin model.

in skin the light include scattering,absorb, and penetrate. that correlation with wavelength

Q3:line 128-129 Human skin consists of two main layers,the superficial epidermal layer and the deep dermal  layer. please explain the phenomenon of light on PWSs.

Reviewer 2 Report

In the paper, the authors proposed a curvature-corrected pressure damage model to predict the rupture of a real radiant vessel. Their work paves a way to provide personalized and precise guidance for clinical treatments. To verify the feasibility of the model, the authors did solid research including simulation and experiments. However, I still have some suggestions for the authors to improve the manuscript:

(1) The authors should clarify the technology they used to capture the flood vessel images in the experiments, such as Figs. 5(a) and 6 (a).

(2) Since the vessel is buried at a depth of ~0.5 mm, the scattering effect of the skin needs to be considered. I suggest the authors cite some references about light scattering in tissue [Appl. Phys. Lett. 111, 201108 (2017), Optica 6(3), 250-256 (2019)] and discuss its influences for their established model.

(3) The authors should update Fig. 5(d) with English characters.

Round 2

Reviewer 1 Report

Acceptable after revision